# Voices from the Ground: Community Perspectives on Preventing Unintentional Child Injuries in Low-Income Settings

**DOI:** 10.3390/ijerph21030272

**Published:** 2024-02-27

**Authors:** Prasanthi Puvanachandra, Anthony Mugeere, Charles Ssemugabo, Olive Kobusingye, Margaret Peden

**Affiliations:** 1The George Institute for Global Health, UK, Imperial College London, London W12 7RZ, UK; mpeden@georgeinstitute.org.uk; 2School of Public Health, Imperial College London, London W12 7RZ, UK; 3School of Population Health, University of New South Wales, Sydney, NSW 2052, Australia; 4Makerere School of Public Health, Kampala P.O. Box 7072, Uganda; mugeere2010@gmail.com (A.M.); cssemugabo@gmail.com (C.S.); theonlyolive@gmail.com (O.K.)

**Keywords:** child unintentional injury, low- and middle-income country, home hazards, community perceptions

## Abstract

Unintentional injuries significantly contribute to mortality and morbidity among children under five, with higher prevalence in low- and middle-income countries (LMICs). Deprived communities in these regions face increased injury risks, yet there is limited research on child safety tailored to their unique challenges. To address this gap, we conducted focus group discussions in rural Uganda, involving parents, village health workers, community leaders, teachers, and maids. The objective was to understand community perceptions around child safety and determine what culturally and age-appropriate solutions may work to prevent child injuries. Analysis of discussions from ten focus groups revealed five main themes: injury causes, child development and behavior, adult behavior, environmental factors, and potential safety kit components. Common injuries included falls, burns, drowning, and poisoning, often linked to environmental hazards such as unsafe bunk beds and wet floors. Financial constraints and limited space emerged as cross-cutting issues. Participants suggested educational resources, first aid knowledge, and practical devices like solar lamps as potential solutions. The study presents invaluable insights into child safety in rural Ugandan homes, emphasizing the role of community awareness and engagement in designing effective, accessible interventions. It underscores the importance of context-specific strategies to prevent childhood injuries in similar resource-constrained environments.

## 1. Introduction

Worldwide, more than 270,000 babies and children under 5 years of age lose their lives every year to injuries [1]. A significant proportion of these deaths occur within low- and middle-income countries (LMIC) where injury-related mortality among young children is four to six times higher than in high-income countries (HICs) [2]. While most evidence, particularly from LMIC, focuses on injury mortality, the detrimental impact of non-fatal injuries and subsequent disabilities on the individual, the society and the health care system is far-reaching (measured per 100,000 people) [3].

With a mortality rate of 100.5 per 100,000 children, the Sub-Saharan African region (SSA) has the world’s highest rate of unintentional injury deaths among children aged 1–4 years [4]. Uganda has an under-five child mortality rate of 94 deaths per 1000 live births (14th highest under-five death rate in the world), of which 5% are attributed to injuries [5,6,7]. Whilst limited in number, studies looking at injuries among this population report that the majority are from burns and falls [8]. Unintentional injuries within the home environment are a major cause of preventable death and serious disability particularly among this pre-school cohort due to the increased amounts of time spent at home and the numerous hazards that exist within those environments [9,10,11,12,13].

It is well known that in environments in which deprived communities live, children are exposed to a greater number of hazards, which, in turn, leads to an increased risk of injury [2,14]. Much of the evidence focused on child injury prevention is both quantitative in nature and is weighted towards HIC where interventions such as smoke alarms, child-proof containers and stairgates have been shown to be effective in reducing child injuries [15,16,17,18,19,20,21,22]. There is a paucity of evidence focussing on what safety interventions would be appropriate to the LMIC home setting and importantly on ascertaining what the uptake and barriers to such interventions would be in resource-limited settings [23,24]. Differences in the social, political and economic environments between HIC and LMIC not only shape the risk profile of child unintentional injuries but also the feasibility, acceptability and sustainability of safety interventions. Qualitative studies provide a unique opportunity to both explore the attitudes, behaviors and perceptions of parents, carers and the wider community towards child safety in greater depth and understand the barriers and facilitators to implementing safety interventions within these settings.

This article highlights findings from qualitative work that was conducted as part of a larger mixed-methods study aimed at developing a culturally appropriate child safety kit to reduce unintentional child injuries occurring in the home environment among children under 5 years of age in the rural setting of Jinja, Uganda. The overarching study was steeped in the Community-Based Participatory Research (CBPR) approach connecting research and translating knowledge gained into local practice [25,26,27]. Through actively engaging with the community at multiple points in the research process, CBPR enables communities and end-users to feel a sense of active engagement and involvement, and importantly, allows for better identification and understanding of the complex issues surrounding child safety [26,28]. The specific aim of this qualitative component was to explore the behaviors and practices of parents and carers in a rural setting in Uganda that put children under the age of 5 at risk of injury within the home environment and to understand what may or may not work in a potential child safety kit whilst simultaneously raising awareness within the community and empowering the mothers/carers to be part of a solution.

## 2. Materials and Methods

### 2.1. Study and Setting

This study took place in Jinja, Uganda—a secondary city located on the shores of Lake Victoria and the source of the River Nile. With a mix of both rural and semi-urban dwellings, Jinja was chosen as being representative of urban and rural settlements in Uganda and other low-income settings.

To capture a wider range of stakeholders, focus group discussions (FGDs) were carried out with parents/caretakers of children under 5 years of age, community health workers, local councilors, daycare teachers and maids. The groups were purposefully kept homogenous such that each focus group consisted of only parents/carers or only community health workers, etc. such that participants would not feel potentially threatened or unable to be open about their thoughts and experiences. This was particularly relevant to the two groups consisting of (a) the mothers and (b) the maids who may have felt intimidated by the presence of males or employers in the room, respectively, when discussing the health and well-being of children. The participants were purposively selected in conjunction with advice from healthcare workers and community leaders who helped identify participants who would be able to actively take part in the FGDs. Eligible participants needed to have the cognitive ability to take part in a discussion and understand the nature of the topics being discussed as well as the ability to speak in the local language. For those groups involving mothers or carers, eligible participants needed to have at least one child under the age of 5 in their care. Table 1 shows a breakdown of the groups that participated in the study.

No personally identifying data were collected and all responses from the participants were kept anonymous with participants being given numbers for reference in the transcripts. Participants were reassured that they could leave the FGD at any point and that all responses would be kept anonymous throughout the data analysis and publication process.

Two research assistants (one male and one female) were trained by the study team in focus group methodology and a moderator’s guide was developed for the research assistants to use. In keeping with good practices for FGDs and wanting to keep the questioning as open as possible, the research assistants were trained to loosely follow the interview guides and probe as necessary. The broader topics that were to be covered in the FGDs were kept largely the same for all of the groups for consistency; however, the wording was altered to suit the audience within the group and ensure clarity. Techniques in probing and moderating were taught and the research assistants were given numerous opportunities to practice their moderation and note-taking skills. Probes were developed by the research team to obtain as much context as possible and additional probes for certain FGD groups such as the councilors or the teachers, who were included to elaborate further on aspects of child safety that would be more pertinent to that target group, e.g., asking teachers whether they cover aspects of home safety for children/siblings of younger children or whether they notice that siblings end up needing to miss school to take care of a younger, injured child. Similarly, probes for the local councilors may include questions relating to any policies that may be in place to enhance child safety within the homes or whether there are public infrastructures or community facilities in place to mitigate against injuries within the homes.

For the pilot focus groups, members of the senior research team sat in with the fieldworkers to provide feedback and suggestions to maximize their interactions with the participants. Informed consent was sought and obtained from each participant who took part before the FGD commenced. Each FGD was recorded using a voice recorder, and audio recordings were translated and transcribed immediately afterward by professional translators working with the research team in Uganda with hand-written notes being incorporated into the transcriptions. A member of the research team periodically checked the translation–transcription process by conducting a back-translation of three of the transcripts as an additional layer of quality control. The transcripts (ranging between 15 and 20 pages of A4 paper) were then ‘cleaned ’ by the fieldworkers and members of the senior research team to eliminate typing mistakes, repetitions and material considered irrelevant to the study, e.g., pertaining to intentional injuries.

### 2.2. Data Analysis

NVivo Qualitative Data Analysis Software Version 2 (NVIVO 2018) was used for the data analysis. Thematic analysis was conducted using the six-phase approach of Braun and Clarke [29]. Two authors (PA and AM) familiarized themselves with the data and generated initial codes. Thematic analysis was used as it is ideal for exploratory work where the aim is to produce a rich and detailed understanding of participant experiences [30]. This study utilized a hybrid approach of thematic analysis that incorporated both a data-driven inductive and an analyst-driven deductive approach [31,32]. In the initial deductive thematic analysis phase, previous experience of the researchers together with a thorough reading of the literature in the area was used to map the theoretical content from which initial categories arose. In the second inductive phase, the previous literature was put to one side and two researchers immersed themselves in the raw data and allowed for new themes to emerge directly from the whole data set. Initial codes were generated by looking for repetition, actions, activities, and beliefs, including those that both conflicted or agreed with each other [33,34]. As themes emerged and developed, they were reviewed in an iterative process against the wider transcript, with other members of the wider research team (CS) and the fieldworkers until a clear definition for each theme was reached. Codebooks and word clouds were used to: identify relationships across codes, and between the levels of codes, i.e., categories, main themes and sub-themes [33,34]. This visualization helped to feed the results back to the participants and share the insights, thereby fulfilling further the CBPR approach to the study.

## 3. Results

A total of ten (10) FGDs were conducted in March 2020 (prior to Uganda going into lockdown) by two research assistants (RAs) who took on alternate roles of moderating and note-taking during the discussions. The 10 groups consisted of mothers (3), maids (2), local councilors (2), teachers (1), parents (1) and village health workers (1). A total of 94 participants were involved in the FGDs.

Five main themes were identified through thematic analysis: cause of injury, child development and behavior, adult behavior, environmental factors and child safety kit elements. The elicited themes, sub-themes and codes that were agreed upon by the research team are shown in Table 2. It is important to note here that many of these themes are cross-cutting in nature and therefore are entwined with each other. Cross-cutting sub-themes (italics in the table) included: blame, supervision, roles and responsibilities, curiosity, and awareness (or lack thereof). These are mentioned throughout the results section. 

### 3.1. Nature of Injury

Perceptions of what constituted an unintentional injury, e.g., a fall, burns, or poisoning, were limited at first to mainly falls and burns; however, the moderator was able to help expand the participants’ knowledge and gain more insight into the other types of injury. In many instances, it was clear that this was the first time that participants had thought about injuries as being a serious threat to their child that was preventable.

#### 3.1.1. Falls

Falls were the most cited cause of injury by the participants, predominantly falls off bunk beds (with and without guard rails) with either two or three decks. Parents recalled instances where children were left to sleep on top decks, and when left unattended or whilst the parents themselves slept, the child would fall and injure themselves. The presence of guard rails was perceived as being of no significant help in preventing falls and in some instances thought to be a hazard in themselves due to their sharp edges.

Some participants complained that the designs of the beds were inappropriate for children but that due to limited space, there was no other choice, meaning that the parents were forced to expose their children to the risks associated with sleeping on high bunk beds:


*“You can decide to sleep with the child on a what…...adults on a double deck bed? So I may forget that I left a child on the top and he can easily fall because deckers are never low, they are always high...too high”*
—Respondent 8, FGD 4

Falls into pit latrines were another commonly cited cause of injury amongst this particular age group whilst playing, often unsupervised, in communal courtyards, which are considered part of the outdoor home environment.


*“In communities where we live you find that there are uncovered holes/pits that are not easy to spot that were dug—the child could be walking and this child falls down and badly”*
—Respondent 2, FGD 6

Other causes of falls, such as slipping on wet floors or off verandas with no rails, were mentioned by some participants. The latter was highlighted by one respondent as being a result of the mimicking nature of young children who do not understand their own limitations as compared to older children:


*“He sees the older kids who jump and go so he also wants to do it to see if he can also jump and I saw him but even that jump resulted ina fracture”*
—Respondent 3, FGD 1

#### 3.1.2. Burns

Burn injuries were mentioned in all of the FGDs as being a common and long-lasting injury sustained by under-fives. The most commonly cited cause of a burn injury was as a result of stoves with children attempting to touch the hot charcoal in an attempt to copy their mothers or the maids:


*“ …maybe us, the women who are used to it—we don’t get burnt by fire and you get hot charcoal from this charcoal stove to another and now this child sees you and also wants to pick the fire charcoal up and also transfer it the way we do and this fire burns the child”*
—Respondent 4, FGD 1

Another commonly cited reason for children burning themselves on hot stoves was due to children being hungry and unsupervised. It was apparent that it is common practice in these crowded households to have a continuously cooking pot of porridge often left unattended.

These instances reflect other cross-cutting themes of mimicry (child behavior) and supervision (adult behavior) which are discussed later. There were several other causes of burn injuries that participants mentioned, some of which were naturally upsetting to hear within the group and elicited heightened emotion among the participants. Fires caused by unattended lit candles and paraffin lamps, often in the sleeping areas, were mentioned by a few participants:


*“…we call it “tadoba”—a light with paraffin with the help of a thread and when you are doing something maybe outside and the children are there inside but sometimes the curtain is also near the wind blows it and everything burns, the house will burn and the child will burn.”*
—Respondent 6, FGD8

Negligence and lack of awareness of the risks to the child were mentioned in several cases where fires had resulted in devastating consequences for a child:


*“I had a friend here in Mbikko, she left them and power went off and she lit a candle she had gone to have fun. At 3:00am a nine months old baby, got burnt and died in the house because of candles.”*
—Respondent 4, FGD 4

In one instance, a respondent linked the requirement to have paraffin lamps around the house with the lack of financial ability to afford electricity:


*“You see the power may go off and you can’t afford for it to turn on again and then you buy a candle……There is a baby in the bedroom, you have put another candle in the sitting room and you are outside cooking. The baby can be easily burnt.”*
—Respondent 5, FGD 2

A few of the participants mentioned other causes of burns, namely electrical burns due to faulty wiring or exposed powerlines and burns resulting from touching the hot engines of the “bodabodas” (motorcycles) parked in the communal courtyards.

#### 3.1.3. Drowning

Drowning injuries were mainly seen as a result of unattended buckets of water or uncovered water drums in the courtyard.


*“Another thing that we do and it causes children risks of injuries, you may collect rain water in a drum and then leave it there. So when a child comes to look in the drum, it is the head that goes into the drum first.”*
—Respondent 9, FGD 4

Although some of the participants did mention unfenced swimming pools, it was apparent that these instances were based on the theoretical ideas of some participants and were not the reality faced by the community. In fact, the moderators noted the ripple of laughter around the room when this was mentioned as a possible cause of drowning. However, it did then bring about discussion of access to larger bodies of water such as the lakes and riversides where many of these communities reside:


**
*“*
**
*Okay for us we are next to the lake, so it is easy for a child to go to the lake. We always get such problems of children falling in water and they drown.”*
—Respondent 5, FGD 4

#### 3.1.4. Poisoning

Hazards surrounding the unsafe storage of various poisonous substances were mentioned in several FGDs. Participants commented on the “negligent” practices of households leaving common, everyday substances within reach of young children:


*“Some parents, we buy booze and take it around children. So when you take that booze or say you have finished the bottle, a child between the ages of one two years is naïve and the paraffin you also put it in a what……in a bottle. A child may come thinking it is taking what dad or mum always takes but is drinking paraffin”*
—Respondent 8,. FGD 4


*“The problems we normally get with our child of that age bracket, we parents are negligent about poisonous objects like paraffin or rat poison, it doesn’t know the good and the bad, it just picks and drinks. So if you a parent you may rush and give that child first aid, by giving him milk so that it may neutralize the poisonous substance and then rush him or her to the health facility if you can afford that…”*
—Respondent 1, FGD 4

This latter quote highlights several aspects surrounding child safety such as child development and cognition as well as parental education on basic first aid training. It also touches upon the sub-theme of financial constraints limiting access to healthcare for some families.

The notion that certain substances such as alcohol or drugs were also commonly found to be within reach of children resulted in some often-heated discussions surrounding the interweaving sub-theme of “blame” and “negligence”. Parents and maids, in particular, were blamed for carelessly leaving toxic substances around the house within reach of the child leading to severe injuries.

Some interesting comments revolved around the issue of illiteracy with mothers not being able to understand the correct dose of medicine to give children, e.g., antihistamines resulting in overdosing. Another more unusual cause of poisoning that was brought up in a few of the FGDs was that of cement poisoning related to the lack of yard hygiene. It was clear that there are a lot of builders within the communities that we were studying and the practice of storing (sometimes open) cement bags in the courtyards was common:


*“….cement is another one……especially parents who are builders or even yourself you can buy cement you want to repair something, you leave that cement there. A child comes, they normally eat the soil so it may think that it is soil and the child eats the cement and is poisoned but there is so little you can do to help them when this has happened…we cannot always afford to go to the hospital and the village doctors are usually not around for this sort of thing…”*
—Respondent 3, FGD 9

This quote is another example of where a lack of first aid knowledge and barriers to accessing healthcare were apparent.

#### 3.1.5. Other Causes of Injury

Whilst most of the injuries that were mentioned fell into the aforementioned categories, there were a handful of other causes that participants shared. Injuries due to sharp objects such as knives or tools in the courtyard were reported with poor yard hygiene being cited again as reasons why children can access these items. The lack of “slashing” the courtyards, or cleaning them and taking away rubbish/debris, was mentioned as a reason why children would obtain access to sharp tools and broken glass in the courtyard but also as a reason as to why dangerous animals such as snakes would be inhabiting the courtyard resulting in snake bite injuries:

### 3.2. Child Development and Behaviour

The theme of child development and behavior was a common thread throughout all FGDs when the participants were asked to explore the reasons why and how children under five become injured. A child’s natural curiosity was often cited as a reason as to why they find themselves in precarious positions which could lead to fall, burn or poisoning injuries:


*“The children are not safe because we are not with them. These children want so much to explore, you can tell this child that don’t touch that socket but since s/he sees you touching it everyday s/he will touch it immediately after seeing you off. He wants to try and see what his father normally does. So, I say that these children are not safe because they want to discover what happens when I put in that socket or solar and they end up when they have got injuries where we are not. Just because they want to explore.”*
—Respondent 4, FGD 10

The sub-theme of mimicry was a recurring topic in the conversations. Participants acknowledged that children of this age imitate their parents or older siblings, putting themselves in danger. Examples included copying adults by attempting to handle hot charcoal or drinking from beer bottles filled with paraffin.

The lack of toys and playing equipment led to children becoming bored and exploring more. Participants shared incidents where children, left unattended while thought to be playing in the bedroom, wandered off and fell from a veranda. This was attributed to financial constraints and the inability to afford toys. It also highlighted the issue of “knowledge” among both children and parents/caretakers. Debates arose in the FGDs regarding whether children of this age could distinguish between safe and unsafe practices. Some blamed the child for not considering the consequences, while others blamed the responsible adult for not understanding the child’s mindset and behavior:


*“You cannot expect a child to know what is good, what is bad, that is our job as parents and I think that we are partly the cause, us adults, we don’t teach them what is right, what is wrong—why? Because we think they know already? But these children—they are too small to understand…”*
—Respondent 6, FGD 10

Participants talked about the inability to afford childcare and therefore needing to rely on family members to help take care of young children. Parents recalled experiences of where they needed to leave their young children with their older offspring which led to an injury event—often mentioning that they either realized it was inappropriate or that the older sibling should have been more responsible, highlighting the blame culture surrounding child injuries.


*“Like me, I have left a child of 6 months with a child of ten years—just because she is joining senior does not mean that she is old enough. My child it got a bad cut but the older one didn’t do anything-she doesn’t know what to do with this baby but I have no choice have I? I feel bad but I have to work too or else what? No food? No house?”*
—Respondent 7, FGD 1

### 3.3. Adult Behaviour

Much of the commentary that fell within this overarching theme revolved around supervision practices and awareness. Participants often mentioned that children are injured more when the mother or the maid is too busy doing many other chores or jobs elsewhere, sometimes outside of the house and therefore the young children are not adequately supervised and exposed to higher risks. The cross-cutting themes of poverty and lack of awareness are also present in this following quote which highlights the plight of many of the families living in such conditions:


*“At my place there is a lady who locks the child in the house to go work… One time the child was locked inside, the neighbour’s house caught fire when the children were locked inside and now these children for example this young one saw the fire but had no idea what to do…Eventually somebody went in and reached for the child and got him out but these children were already hurt by fire. So, I would also say that being poor is also one of those that lead to these children to get injuries.”*
—Respondent 8, FGD 10

### 3.4. Social Determinants

The theme of social determinants, including financial constraints, limited space, poverty, and education levels of parents/caretakers, is intricately linked to previously discussed themes and sub-themes. Discussions revealed that these communities face significant hardships, living in cramped conditions where bedrooms, living spaces, and kitchens are all in the same small area. These conditions have a profound impact on families, making it challenging to address them with a child safety kit, as mentioned by a focus group participant when discussing prevention solutions:


*“This is hard…I cannot think of things because there is prevention that needs like big things which will not fit in the box…maybe something that will afford us bigger spaces or allow the parent to go and work without worrying…what solution like that can fit in a box? [laughter]”*
—Respondent 8, FGD 10

### 3.5. Safety Kit Elements

A large proportion of the conversation content surrounding the child safety kit was focused on sensitizing parents/carers to child safety as a concept and increasing their awareness of what hazards existed within their households and what measures could be taken to help prevent injuries. As can be seen from Figure 1, a thematic analysis word cloud developed from the transcripts, “education” and “first aid knowledge” were prominent in the offered solutions. Several participants also suggested that there should be some education around child development so that parents/carers could understand what actions children at different ages should and should not be able to carry out.

Initially, many participants emphasized the importance of injury treatment and recommended including first aid items like plasters and wound-cleaning spirits. It was widely agreed that knowledge on treating specific injuries such as burns or poisoning was limited. Given the limited access to health facilities, knowing immediate actions to take after an injury was deemed valuable. A participant proposed training village leaders, who often serve as the first point of contact in case of injuries, as a sensible approach.

When exploring potential devices that could be included in a kit, participants mainly focused on trying to prevent burn injuries and suggested items such as solar lamps, and barriers to stopping children from accessing hot stoves. Figure 1 highlights that bath thermometers and solar lamps were commonly suggested devices to include in the kit. In addition to burn injury prevention, participants also suggested that covering pit latrines would go a long way to helping prevent fall injuries; however, the associated costs of having to carry this out would be prohibitive:


*“Then we are just concentrating the homes which are in towns but once you go in the villages we have pit-latrine toilets. You know those days we could keep them covered but these days people have left that idea of covering toilets. They leave them open and there are those which are too big whereby a child can even fall inside—we need to educate people to start covering them again—make it a habit. But then covering costs money so that is going to be hard to do.”*
—Respondent 6, FGD 10

Other fall injury solutions that were offered included rails for beds, anti-slip mats for bathrooms, stair gates and toddler walkers/strollers. The prevention of drowning through using basins with plug holes was mentioned and for poisoning prevention, a variety of child-safe locks for cupboards, safety caps for bottles, and lockable storage units for chemicals and medicines were all suggested.

A few of the participants picked up on the mimicking nature of children and suggested that one way to tackle this, and to help alleviate boredom among the younger children, would be to provide role-play toys that were safe to use, e.g., play knives and food, and toy stoves with pots and pans and to accompany those toys with messaging that would teach the children about the potential hazards of using such items.

Overall, the participants found coming up with solutions to prevent injuries among under-fives a difficult task. There were many periods of silence whilst they thought about the topic and often the moderator had to intervene to give them some ideas of the sorts of interventions that could work. This highlights not only the limited awareness of community members of child safety and injury prevention but also the somewhat Herculean task of trying to “fix” much larger problems such as financial difficulties or limited space.

## 4. Discussion

Through a series of ten qualitative focus groups, this study presents a wealth of information relating to unintentional injuries among under-fives in the home setting in rural Uganda and importantly, in their own words, the perceptions and opinions of approximately 100 participants—a viewpoint that is often overlooked in the current literature.

It was clear from the number of times that the moderator had to intervene to bring the discussions back on track and to reexplain definitions abd that the general awareness among the community about unintentional injuries within the home environment was limited. The tendency to want to focus more on other childhood diseases and even intentional injuries indicates how low unintentional injuries are on their priorities. In discussion with the research assistants who conducted the focus groups in the local language, it was apparent that this lack of understanding of the exact meaning of the topic led to frustration and confusion, and therefore, contextualizing the problem and ensuring that the participants were given time to explore the definitions of terminology being discussed was crucial in enabling a more fruitful and relevant conversation. This highlights the important aspect of language barriers and differing cultural interpretations of health, thereby adding strength to the critical role that qualitative studies play in understanding the local culture and social norms [35]. Despite this, there is an indication that parents and other caregivers were aware of some of the hazards that children of that age face within the home, but that they lacked the tools to be able to deal with it, predominantly in the form of education. This was highlighted by the word cloud, which showed the prominence of certain key terms such as education and co-sleeping and is supported by other studies from the area of parental perceptions [36].

Several of the sub-themes and codes that emanated from the thematic analysis were deemed to be “new” areas of focus for the research team as the causes of injury and the hazards mentioned in the FGDs had not emerged during previous quantitative studies, that had formed part of the larger mixed-methods project. Whilst findings from the qualitative study mirrored the leading causes of injury at home from the quantitative surveys and risk assessments that had been carried out, the more contextual data that were elucidated from the FGDs demonstrated that falls into pit latrines and burns from touching hot charcoal (rather than the hot pots themselves) were of significance in this study population—two hazards which had not come out of the quantitative household survey or in other studies from Uganda looking at child injuries [8,37,38]. This lays credence to the mixed deductive–inductive approach to thematic analysis that was undertaken in this study [39].

The themes of child and adult behavior that arose from the FGDs are key to understanding what the parents/carers and other community members perceive to be the reasons for why and how children are injured which, in turn, frames how a potential child safety kit may or may not be utilized. For example, the cross-cutting theme of child development and curiosity highlighted an overall agreement that children are curious by nature and will explore without fear of consequence. There was a sense of fatalism to some of the comments indicating that some participants believed that there was no way to prevent such injuries and that there was no way to “control a child’s natural curiosity” so they would inevitably end up injuring themselves. Such beliefs would need to be addressed effectively as part of an overall education and awareness-raising component of a safety kit. Moving the onus away from “blaming the child” to one of “taking responsibility” and “altering the environment” would need to be a fundamental aspect of any package that was developed. Helping adults understand the cognitive abilities of children at various ages and how they interpret the environment around them would be beneficial to help implement any intervention [40].

In a similar manner, the notion that adults were negligent or that there was a lack of supervision and responsibility arose as a common belief among participants. Some participants were adamant that their financial or social situation would not lend itself to changing the situation or alleviating the lack of supervision; however, others acknowledged the role that parents/carers could have in injury prevention. Similar findings were highlighted in another study from Uganda looking at caregiver perceptions in rural settings where financial restraints and other social and environmental factors were identified as issues leading to increased injuries among children [41]. The limited ability of caregivers to supervise children in many rural low-income settings has been well documented and is often linked to poverty, unemployment and poor working conditions [42,43]. Evidence indicates that the likelihood of unintentional injuries among children is three times higher among unsupervised children when compared to supervised children [44]. The lack of affordable quality childcare was mentioned by participants in our study and is mirrored in other parts of Africa such as South Africa and Botswana where studies reveal that caregivers are forced to leave children unattended or under the supervision of siblings [45,46].

With regard to prevention, it was highly apparent from the overall themes that emerged and from the more specific discussions relating to prevention that whilst certain elements such as solar lamps instead of paraffin lamps and safety caps or bath thermometers may be included in a “kit”, there were much broader issues revolving around determinants that would not be as easy to solve through the provision of a kit. It was clear, as supported by similar studies in other countries, that increasing the awareness of parents and carers alike through education materials and first aid training would be beneficial and that particular cultural practices such as co-sleeping would need to be addressed [47,48]. This complex interplay between safety devices and social determinants necessitates a holistic approach to preventing injuries in this population. While the development of a child safety kit represents a tangible step towards mitigating specific hazards, its effectiveness is inherently linked to broader efforts aimed at enhancing community awareness, education, and the modification of cultural practices.

This study highlights the perceptions of mothers, caregivers, and the wider community on unintentional injuries in children under five. Capturing diverse perspectives is critical for system-wide approaches to injury reduction. However, the study had limitations. Focus groups carry inherent risks, including participants feeling unable to voice their own perceptions due to hierarchies and social contexts. Efforts were made to mitigate this by maintaining homogeneity of groups, but some views may have remained unheard. The study was limited to rural Uganda to capture relevant perceptions and factors, which may affect generalizability to urban areas of Uganda and other parts of Sub-Saharan Africa.

## 5. Conclusions

This qualitative work, along with the quantitative components of the larger study, provides critical insight into behavioral, environmental, and social factors related to home injuries. The findings suggest a need to increase awareness, skills, and knowledge of caregivers and the community regarding unintentional injuries in children under five. By engaging community members in the research process and asking them to help develop solutions, we fostered a sense of ownership and empowerment, enabling them to identify context-specific interventions and contribute to the development of effective prevention strategies. Participants suggested solutions such as providing educational materials, first aid, and home hazard reduction to alleviate the burden of injuries. Our findings have the potential to inform policymakers, healthcare providers, and community-based organizations in designing targeted interventions that resonate with the realities and needs of the communities they serve. Future work should focus on developing such materials and involving the wider community in piloting and testing potential interventions and packages that are affordable and accessible to rural communities within Uganda.

## Figures and Tables

**Figure 1 ijerph-21-00272-f001:**
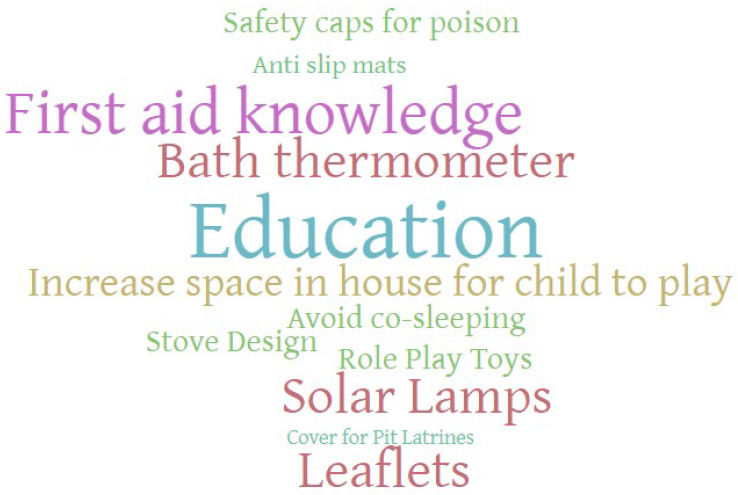
Thematic analysis word cloud for child safety kit elements—all focus groups combined.

**Table 1 ijerph-21-00272-t001:** Characteristics of Focus Group Participants.

Focus Group	Population	Description	Age Range	Number of Participants	Location
1	Maids	Employees who were specifically paid to look after at least one child under the age of 5 within that household (all female)	15–29	10	Kakindu
2	Maids	Employees who were specifically paid to look after at least one child under the age of 5 within that household (all female)	18–34	10	Mpumudde
3	Mothers of children < 5 years	A mother who had at least one child under the age of 5 in a household (all female)	19–33	10	Kakindu
4	Mothers of children < 5 years	A mother who had at least one child under the age of 5 in a household (all female)	22–45	10	Mpumudde
5	Mothers of children < 5 years	A mother who had at least one child under the age of 5 in a household (all female)	27–50	8	Walukuba
6	Local Councillors	Elected officials who represent the interests of residents within specific wards, divisions, or municipalities in Jinja (mixed men and women)	36–64	8	Jinja Municipality Council Chambers
7	Local Councillors	Elected officials who represent the interests of residents within specific wards, divisions, or municipalities in Jinja (mixed men and women).	39–59	10	Kakindu parish
8	Teachers	Teachers employed in early childcare settings in Jinja teaching pre-school age children (i.e., under-five (mixed men and women).	25–38	9	Kakindu Community Centre
9	Parents (mixed)	A group of 6 men and 4 women who were all parents of at least one child under 5 years (mixed men and women).	28–46	10	Walukuba Health Centre IV
10	Village Health Workers	Community-based health workers who act as the first point of contact for health-related issues within the village (mixed men and women).	27–48	9	Kakindu Community Centre

**Table 2 ijerph-21-00272-t002:** Themes, subthemes and codes elicited by thematic analysis.

Main Theme	Sub-Themes	Codes
Cause of injury	Falls	Off beds
		Pit latrines and other pits/holes
		Verandas
		Slips in bathrooms
	Burns	Hot stoves
		Water
		Fire
		Boda-boda engines
		Electrical burns
	Drowning	Bath basins Swimming Pools/Lakes/riversides Latrine pits/outdoor water drums
	Blunt/sharp injury	Knife injuries Falling fruit from trees Play with other children Broken bottles
	Poisoning	Chemicals including paraffin Alcohol Drugs Cement Indoor flowers
	Animal bites	Dog/Cat bites Snake bites
	Choking	Small toys Beads for hair
Child Development and Behaviour	Curiosity	Exploration Mimicry Boredom
	Supervision and responsibility by older siblings	Chores Knowledge of child/sibling Caretaker role if parents work
Adult behavior	Supervision and responsibility	Numerous chores
		Care seeking behavior
		Job-seeking
	Awareness	Knowledge and education
Social Determinants	Poverty	Space
		Finance
	Access to healthcare	Care-seeking behavior
		Knowledge and awareness
Elements to consider for child safety kit	Devices	Burns
		Falls
		Drowning
		Poisoning
		Cuts/blunt injuries
		Mimicking nature
	Education Component	Sensitization
		Environmental Factors

## Data Availability

Data are contained within the article.

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
