# Peer review of "Voices from the Ground: Community Perspectives on Preventing Unintentional Child Injuries in Low-Income Settings"

_ijerph, 2024, doi:10.3390/ijerph21030272_

Round 1

Reviewer 1 Report

Comments and Suggestions for Authors

This manuscript offers qualitative data from rural Uganda concerning unintentional injury risk among children under age 5. The manuscript is nicely written and organized, and the data contribute to the small and understudied area of research on unintentional pediatric injuries in LMIC, including Uganda. I have several comments for the authors to consider.

1.      It would be nice to have more details on the participants in the focus groups. I have several questions immediately, and other details might be provided also:

a.       How many people were in each group (in the discussion, it mentions 100 participants total – was that 10 members of each of the 10 focus groups?)?

b.      What were the age and sex mixes in the groups?

c.       Were parents/caretakers mostly parents, or were there several grandparents, older siblings, or even non-family members?

d.      Who were the policymakers and how are they different from community leaders?

e.       What is meant by “maids” – are these people who clean homes (my initial interpretation of the term ‘maid’ from an American perspective but I suspect not what is intended), who watch children (like a nanny or babysitter), or something else?

f.        Did participants provide informed consent? I’m guessing they probably did, but this detail should be explicitly mentioned.

2.      Any information on the accuracy of the translation of the transcripts? Translation is not simple, especially with culturally-sensitive topics like this one.

3.      I’d encourage the authors to review concordance between Table 1 and the text. In the drowning section, for example, the table seems to emphasize slightly different risks than section 3.1.3; the notion of natural bodies of water such as lakes and rivers is not even mentioned in the table but is prominent in the text.

4.      The quote from Respondent 8, FGD 4 concerning booze and paraffin appears twice in the article, and probably should be used only once.

5.      I’d encourage the authors to update their literature review. Very few recent articles are cited (in fact, I believe there are no citations from after 2019 except the author’s own 2022 paper in reference 37). I worry the authors simply used citations from their grant proposal without updating the literature, which of course is a serious problem and poor scientific practice. As a quick check, I typed “child injury rural Uganda” into Google Scholar, limited the search to publications since 2020, and found at least 3-4 articles that seem highly relevant and should be incorporated into this manuscript’s introduction and discussion section.

6.      I disagree with the author’s statement concerning data availability and data sharing. They have rich qualitative data that could be anonymized and made available to interested and qualified fellow scientists who want to replicate and/or extend the current findings. I’d encourage them to make the data available upon request, unless there are ethical prohibitions I am not thinking about.

Reviewer 2 Report

Comments and Suggestions for Authors

The study represents a novel and qualitative approach to understanding child injury and potential injury prevention solutions, using focus groups conducted in the native language with mothers and caregivers.  I found it interesting and useful for local planning and intervention purposes.   All authors have strong backgrounds in the field and most have local knowledge and have been prolific publishers in injury prevention and control.

Line (l) 62     The article states that’s its purpose was to develop a culturally appropriate child safety “KIT”, through conducting focus groups of mothers and caregivers.   In the Discussion and conclusions, however, they do not follow-up or come back to the KIT with more information on how the study has or would have led to the kit they describe in Fig 1.   I believe the intent was more to understand local knowledge and risks to children in order to know  local perceptions, beliefs, and barriers to preventing injury. Here they describe studying “injuries in the home yet later (l 166 and 265) they use the phrase “and in common courtyards”   I suggest adding that phrase to line 62.

L 64    I believe the correct title of the approach is Community-Based Participatory Research. (CBPR)

L 81-85     as per line 368 , mention here in Methods, that for your focus groups,  had 10 groups of about 10 persons each.  How long did the focus group(s) last, each? Were men and women represented equally?   What was the age structure?  What were the age range  of the children they take care of?

L 82.   What do you mean by groups were kept separate?    From what or whom?   I assume you mean you did not have one large group….were all groups queried at the same time or at different times, over what stretch of time (perhaps to control for seasonal differences in environments which may bias their answers).     What was the time frame these were conducted ? (2018? 2019?).

Can you elucidate a bit more about what the script for the focus groups looked like?    That is, what kinds of questions were posed, were the same questions used for all groups?  Were the research assistants conducting the FG about the same age?   All women interviewers?  (e.g. did you ask “what would you put into a kit to give to families to prevent these injuries?”)

Table one…under poisoning, there is no mention of paraffin, which you note later as a comment from respondants.

There is not much here about the KIT….not sure figure one does justice to your thinking about a Kit, and how it would materialize from the results.     From Fig 1, “increase space in house for child to play”, “avoid co sleeping” do not seem part of a kit (with solar lamps, etc), so maybe that goes in as recommendations in the kit, or “myths and facts” sheet in the kit.

 References

 25-26-27-28 have noted (in eng) after title.   Why?   No other American journals listed have that noted.   I would omit those notes.

30-31-33 have “sage” as publisher instead of Sage (as in 34)
